# AnalogGenie-Lite: Enhancing Scalability and Precision in Circuit Topology Discovery through Lightweight Graph Modeling

Jian Gao [1]   Weidong Cao [2]   Xuan Zhang [1]

## Abstract

The sustainable performance improvements of integrated circuits (ICs) drive the continuous advancement of nearly all transformative technologies. Since its invention, IC performance enhancements have been dominated by scaling the semiconductor technology. Yet, as Moore's law tapers off, a crucial question arises: ***How can we sustain IC performance in the post-Moore era?*** Creating new circuit topologies has emerged as a promising pathway to address this fundamental need. This work proposes AnalogGenie-Lite, a decoder-only transformer that discovers novel analog IC topologies with significantly enhanced scalability and precision via lightweight graph modeling. AnalogGenie-Lite makes several unique contributions, including concise device-pin representations (i.e., advancing the best prior art from $O\left(n^2\right)$ to $O\left(n\right)$), frequent sub-graph mining, and optimal sequence modeling. Compared to state-of-the-art circuit topology discovery methods, it achieves $5.15\times$ to $71.11\times$ gains in scalability and 23.5% to 33.6% improvements in validity. Case studies on other domains' graphs are also provided to show the broader applicability of the proposed graph modeling approach. Source code: https://github.com/xz-group/AnalogGenie-Lite.

## 1. Introduction

Nearly every transformative technology over the past several decades, ranging from medical devices and 5G communication to generative AI and quantum computing, has been pow-

ered fundamentally by the semiconductor integrated circuits (ICs) technology. The relentless performance improvements of ICs have been the driving force behind the continuous advancement of these innovations and will continue to play a key role in enabling future technological breakthroughs. Historically, remarkable progress in IC performance has been primarily driven by semiconductor process scaling, evolving from 10 $\mu$m in the early 1960s to the cutting-edge 2 nm nodes of today – a $5000\times$ in dimension reduction and nearly $100000\times$ in performance improvement. This progression laid the foundation for the well-known Moore's law, which predicts that computing power (i.e., CPU performance) doubles approximately every 18 months – a principle that has shaped the trajectory of modern technology. Yet, the end of Moore's law is looming (Theis & Wong, 2017) and the continuous performance enhancements of ICs can no longer be expected from technology scaling, a crucial question thus arises: ***How can we sustain and advance IC performance in the post-Moore era?***

Developing novel circuit topologies, especially for analog ICs, has emerged as a promising pathway to meet this critical challenge (Schuman et al., 2022; Mohseni et al., 2022; De Leon et al., 2021). Analog ICs essentially bridge the physical world and cyberspace by dealing with continuous signals and enabling seamless interaction with digital ICs in the cyber domain that handle binary data. Their topologies (i.e., interconnections between different devices) primarily determine the performance, functionality, and efficiency of a circuit once a semiconductor process is finalized to implement the circuit, akin to a protein structure that dictates the generation of amino acids. Yet, unlike their digital counterparts that can be easily synthesized with high-level hardware description languages (e.g., Verilog and VHDL) or programming languages (e.g., C) or even multiple generative AI-based tools (Blocklove et al., 2023; Thakur et al., 2024; Fu et al., 2023; Wu et al., 2024; Liu et al., 2023), analog ICs have relied on a longstanding handcraft design process. This is mainly due to their inherent complexity, which makes them resistant to the universal and hierarchical abstraction that digital ICs benefit from. Thus, the discovery of novel analog circuit topologies has been significantly limited since the invention of ICs and remains a critical challenge to be addressed.

[1]Department of Electrical and Computer Engineering, Northeastern University, Boston, MA, USA [2]Department of Electrical and Computer Engineering, The George Washington University, NW Washington, DC, USA. Correspondence to: Xuan Zhang <xuan.zhang@northeastern.edu>.

*Proceedings of the $42^{nd}$ International Conference on Machine Learning*, Vancouver, Canada. PMLR 267, 2025. Copyright 2025 by the author(s).

Generative AI has recently emerged as a promising solution for automating analog circuit topology discovery. To facilitate the discovery, pioneering methods have explored two typical representations to model analog circuit topologies, i.e., text (Lai et al., 2024; Chen et al., 2024) and graph (Dong et al., 2023; Chang et al., 2024; Gao et al., 2025). Text representation uses PySpice codes or natural language to describe analog circuits, which can be converted to a SPICE (Simulation Program with Integrated Circuit Emphasis) netlist – a textual high-level description of device connections used for circuit simulation. Graph representation formulates the topology design as a graph/sequence generation task, as circuit topologies of analog ICs can be naturally represented as graph structures. Yet, these representations face significant limitations that must be addressed to enable precise and scalable discovery of topologies. Text-based representations are prone to errors, as generating even a single device or connection often involves predicting multiple text tokens. Even advanced LLM models such as GPT-4 (Lai et al., 2024) struggle to generate circuits with fewer than 10 devices accurately. While graph-based representations can significantly mitigate errors by predicting device-pin connections, they still suffer from scalability and accuracy. Specifically, these methods (Dong et al., 2023; Chang et al., 2024; Gao et al., 2025) do not efficiently and precisely model circuit graphs. When leveraging auto-regressive models such as transformers that have a limited context window for circuit graph generation, they are unable to generate circuits at scale and are limited to accuracy due to the error propagation of long-sequence predictions (Zhu et al., 2022).

This work proposes AnalogGenie-Lite, a generative engine based on a decoder-only transformer for discovering novel analog circuit topologies with enhanced scalability and precision through lightweight graph modeling. Compared to its predecessor (Gao et al., 2025), AnalogGenie-Lite harnesses a lightweight circuit graph modeling to address the limitations of existing methods. At the graph level, it simplifies the cutting-edge device-pin graph representation (Gao et al., 2025) by removing redundant nodes and edges, reducing graph complexity for multi-pin shared edge connection from $O\left(n^2\right)$ to $O\left(n\right)$. At the subgraph level, it employs data mining on a database (Gao et al., 2025) of more than 3000 topologies across tens of circuit types to identify and simplify frequently reused subcircuits, replacing them with compact representations. Finally, it models a circuit graph as a shortest closed path that visits every edge of an undirected graph at least once by solving the *Chinese Postman Problem* (Edmonds & Johnson, 1973) and significantly reduce sequence length compared to previous non-optimal methods (Gao et al., 2025). These innovations remarkably enhance the scalability and precision of topology discovery. Moreover, this efficient graph modeling method from AnalogGenie-Lite can also be broadly applied to other do-

mains such as protein generation (Jumper et al., 2021) and community detection (Fortunato, 2010). Our key contributions are as follows:

- **Precise and efficient graph modeling**: AnalogGenie-Lite significantly improves the efficiency of graph representation at the device-pin level for analog circuit topology modeling. It exploits opportunities to eliminate redundant device self nodes and simplifies multi-pin shared edge connections, reducing space complexity from $O\left(n^2\right)$ to $O\left(n\right)$.

- **Compact subgraph modeling**: AnalogGenie-Lite employs frequent subgraph mining to identify commonly reused subgraphs in the database. It further simplifies these subgraphs by pruning isolated nodes and restructuring the remaining non-isolated nodes into a cycle.

- **Optimal sequence modeling**: AnalogGenie-Lite models the graph as a shortest closed path that visits every edge of an undirected graph at least once by solving the *Chinese Postman Problem* (Edmonds & Johnson, 1973). This significantly reduces sequence length compared to previous non-optimal methods.

- Experimental results show that AnalogGenie-Lite achieves remarkable generation performance, i.e., $5.15\times$ to $71.11\times$ reduction in average sequence length across the entire database and $23.5\%$ to $33.6\%$ improvement in validity, compared to the best prior art. Case studies on other domains' graph datasets (e.g., protein, ego, community, molecule, and 3D point cloud graphs) further showcase the broad applicability and superior scalability of our graph modeling approach.

## 2. Preliminaries and Related Work

### 2.1. Analog Circuit Design Flow

The analog circuit design process generally involves three key stages. First, it starts with creating the circuit topology, which entails selecting device types (e.g., transistors and resistors), determining the number of devices, and defining their interconnections. Next, designers carry out device sizing, i.e., optimizing the physical dimensions of the devices to achieve specific performance objectives. Finally, the physical layout based on the topology and device dimension is created for fabrication, which represents ICs as stacked physical layers used in manufacturing. While significant advancements have been made in automating the device sizing (Wang et al., 2020; Cao et al., 2022; Gao et al., 2023; Cao et al., 2024) and layout design stages (Kunal et al., 2019; Xu et al., 2019), the problem of topology generation remains significantly underexplored due to its abstract, complex nature, requiring creativity and human-level intelligence. Our work addresses this challenging problem.

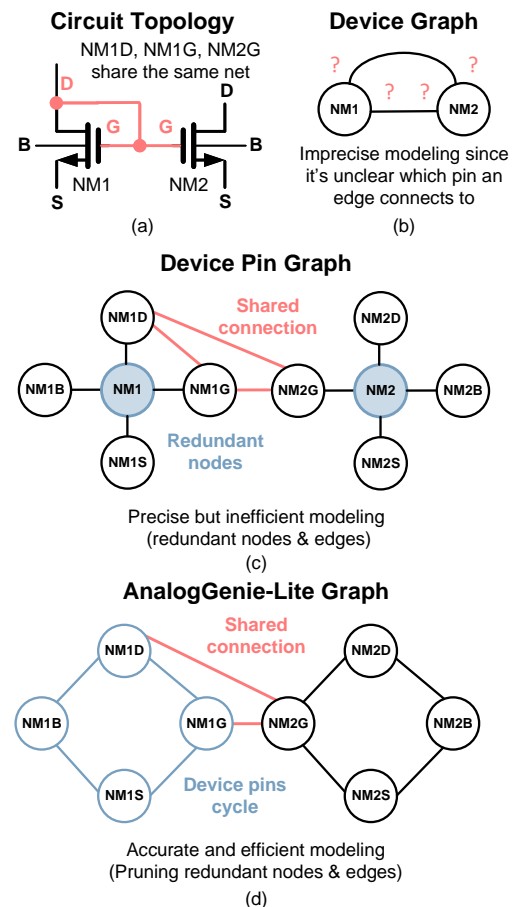

**Circuit Topology**
NM1D, NM1G, NM2G share the same net

(a)

**Device Graph**
Imprecise modeling since it's unclear which pin an edge connects to
(b)

**Device Pin Graph**
Precise but inefficient modeling (redundant nodes & edges)
(c)

**AnalogGenie-Lite Graph**
Accurate and efficient modeling (Pruning redundant nodes & edges)
(d)

*Figure 1.* AnalogGenie-Lite's accurate and efficient graph modeling by pruning redundant nodes and edges from device pin graph.

## 2.2. Existing Analog Circuit Topology Generation

Generative AI has demonstrated significant potential in tackling topology generation, with early approaches focusing on text-based or graph-based methods. For example, Analog-Coder (Lai et al., 2024) models circuits as Python-style SPICE (PySpice) netlists, using domain-specific prompt engineering with large language models (LLMs). While this flexible representation allows diverse and scalable designs, its reliance on high-level text introduces challenges in generating valid circuits due to the need for multi-token predictions for each device or connection. Artisan (Chen et al., 2024) avoids this complexity by focusing on topology selection rather than generation. Using natural language to represent topology names, Artisan reuses existing topologies, avoiding the complexity of low-level design. However, this limits its ability to generate novel and custom topologies. Graph-based methods, such as CktGNN (Dong et al., 2023), use adjacency matrices and a graph variational autoencoder (VAE) to generate topologies for specific analog ICs like operational amplifiers (20 devices). Similarly,

LaMAGIC(Chang et al., 2024) fine-tunes a masked language model (MLM) to generate fixed-node circuits, achieving high success for simpler designs like power converters (fewer than four devices). AnalogGenie (Gao et al., 2025) uses non-optimal Eulerian circuits to model the graph. This efficient representation avoids encoding non-existent edges, enabling scalability to larger circuits (e.g., 63 devices). In this work, AnalogGenie-Lite further advances analog circuit topology generation with lightweight graph modeling.

## 3. Approach

AnalogGenie-Lite is a domain-specific generative model tailored to discover novel analog circuit topologies of versatile types with exceptional scalability and precision. To achieve this, AnalogGenie-Lite leverages three key innovations in its lightweight graph modeling. First, it significantly improves the efficiency of the device-pin graph, which accurately represents circuit topologies. By eliminating redundant nodes and edges, it simplifies the graph structure, enabling more efficient and precise topology discovery. Second, AnalogGenie-Lite employs frequent subgraph mining to identify commonly used subcircuits and introduces a customized tokenizer that incorporates both fundamental device and subcircuit pins. This approach allows for the versatile generation of custom circuits while enabling efficient reuse of existing subcircuits. Third, it models circuit graphs as the shortest closed path visiting all edges at least once by solving the *Chinese Postman Problem* (Edmonds & Johnson, 1973). This technique significantly reduces sequence length compared to non-optimal methods (Gao et al., 2025).

### 3.1. Precise and Efficient Graph Modeling

Abstracting the efficient and precise representation of an analog circuit topology is a long-standing challenge. Graph representations have recently emerged as a promising modeling methodology. Early works (Dong et al., 2023; Lu et al., 2023; Lohn & Colombano, 1999; Mattiussi & Floreano, 2007) rely on high-level graphs to generate circuit topologies, where each node represents a device. Although this approach offers efficient representations, it omits essential low-level device details, resulting in ambiguous generation. Consider an NMOS transistor (NM) with four pins, i.e., drain (D), gate (G), source (S), and body (B) in Figure 1a for illustration. Abstracting the entire device as a single node makes it unclear which pin an edge connects to. On the other hand, AnalogGenie (Gao et al., 2025) represents circuits at the pin level, ensuring a unique mapping between the graph and circuit topology, where every connection is explicitly represented (Figure 1c). Specifically, it represents the topology of an analog circuit as a finite connected undirected graph $\mathcal{G} = (V, E)$, where $V = \{1, 2, \ldots, n\}$ is the node set representing each device pin with $|V| = n$ and

$E \in V \times V$ is the edge set. Yet, this approach inevitably increases graph complexity by introducing more nodes and edges and does not scale well as the number of devices in a circuit grows. AnalogGenie-Lite significantly improves the efficiency and accuracy of this graph modeling by pruning redundant nodes and edges.

**Pruning redundant nodes**: In the device pin-level graph structure (Gao et al., 2025), each device is represented by its pins (e.g., NM1D, NM1G, NM1S, NM1B) and the device node itself (e.g., NM1 for an NMOS). The device node primarily models situations where device pins are connected to themselves or remain unconnected. However, here redundancy exists, as representing the connection between device pins that belong to the same device is not required. As an example, the edge NM1D ↔ NM1G is redundant with NM1D ↔ NM2G in device pin graph (Figure 1c). Consequently, AnalogGenie-Lite removes all device itself nodes from the graph and connects device pins nodes as a cycle (Figure 1d).

**Pruning redundant edges**: When $n$ device pins share the same connection (Figure 1a), the device-pin graph (Gao et al., 2025) represents this case using $\frac{n(n-1)}{2}$ edges, connecting every pair of pins explicitly. For example, if three pins (NM1D, NM1G, and NM2G) share a connection (Figure 1c), existing work uses three edges: NM1D ↔ NM1G, NM1D ↔ NM2G, and NM1G ↔ NM2G. AnalogGenie-Lite simplifies this representation by selecting a single node that does not cause edges overlapping with the device pins cycle and connecting all other nodes to it. Thus, only $n-1$ edges are required. In the example, AnalogGenie-Lite selects NM2G as the node since selecting NM1G or NM1D will result in NM1G ↔ NM1D edge that overlaps with NM1 device pins cycle. As a result, it uses only two edges: NM1D ↔ NM2G and NM1G ↔ NM2G to describe the shared connection. This technique remarkably enhances the scalability, as $n-1$ edges scale linearly compared to the quadratic growth of $\frac{n(n-1)}{2}$ edges. More examples of this edge pruning method can be found in Appendix A.1.

### 3.2. Compact Subgraph Modeling

Analog circuit topologies naturally exhibit a hierarchical structure, combining low-level unique devices essential for novel configurations with high-level subcircuits that are versatile and reusable across various circuits. Yet, existing approaches either focus solely on generating analog circuit topologies at the most fundamental device level from scratch (i.e., device pin), resulting in heavyweight sequential structures (Gao et al., 2025), or operate exclusively at the high level (i.e., subcircuit), limiting their versatility to represent diverse topology structures and restricting them to specific circuit types (Dong et al., 2023). AnalogGenie-Lite bridges this gap by enabling the simultaneous exploration of both custom and hierarchical structures. First, AnalogGenie-Lite

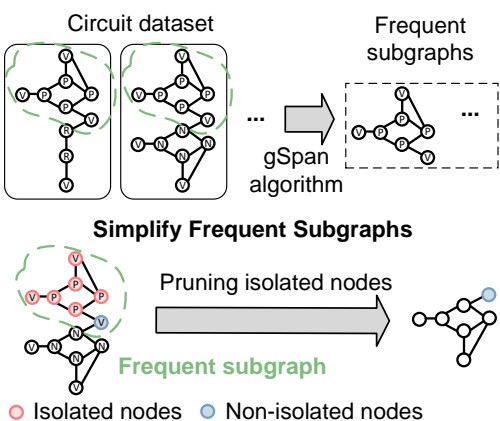

Figure 2. A simplified diagram illustrates how AnalogGenie-Lite identifies and simplifies frequent subgraphs.

employs a domain-specific tokenizer to encode and decode sequences. Each token in AnalogGenie-Lite's tokenizer corresponds to either a device pin (e.g., NM1G, NM1D, NM1S, NM1B) or a circuit-level pin for the overall analog topology (e.g., VIN1, VOUT1, VDD, VSS). This approach allows AnalogGenie-Lite to generate topologies at the most fundamental level by predicting the next device pin token. Additionally, AnalogGenie-Lite incorporates special tokens to represent the pins of subgraphs or SG (e.g., SG1_VDD, SG1_VOUT, SG1_termA, SG1_termB, SG1_termC). These tokens enable the generation of only the outer pins of a subcircuit, omitting inner low-level design details. This significantly simplifies the graph structure while maintaining the ability to represent hierarchical topologies. Detailed tokenizer table is shown in Table 2 in Appendix A.2.

Additionally, existing approaches either construct an extensive design library (i.e., tokenizer table) to significantly simplify the graph structure, which includes numerous infrequent subcircuits and inefficiently utilizes the design library's space (Gielen & Rutenbar, 2000; Zhao & Zhang, 2022; 2020), or rely solely on heuristic design knowledge to identify frequent subcircuits, which may lead to inaccuracies (Dong et al., 2023). AnalogGenie-Lite, on the other hand, employs rigorous data mining on a large analog circuit topology database (Gao et al., 2025) comprising over 3000 unique topologies and 11 types of analog circuits to accurately **identify** the frequent subcircuits that are actively reused within the database (Figure 2). Specifically, AnalogGenie-Lite utilizes the gSpan algorithm (Yan & Han, 2002) to mine frequent subgraphs. gSpan discovers frequently connected subgraphs in a graph dataset without explicitly generating candidate subgraphs, overcoming the limitations of earlier Apriori-based approaches and significantly reducing runtime. Once frequent subgraphs are

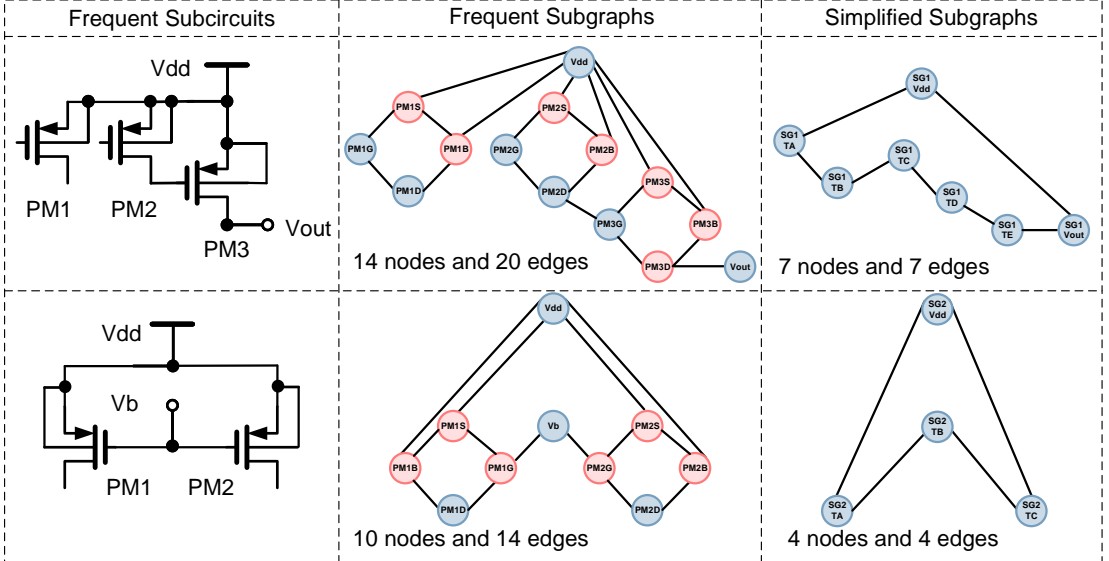

*Figure 3.* Two frequent subgraph examples (over 25% appearing frequency in the dataset and over 50% of nodes within subgraphs are isolated) that AnalogGenie-Lite selects to identify and simplify (red: isolated nodes; blue: non-isolated nodes).

identified, AnalogGenie-Lite classifies the nodes within subgraphs as isolated nodes (nodes connected only to other nodes within the subgraph) or non-isolated nodes (nodes also connected to the rest of the graph) based on their degree. To **simplify** the frequent subgraphs, AnalogGenie-Lite prunes the isolated nodes and their edges (Figure 2). Next, it connects the remaining non-isolated nodes in a cycle and replaces the original subgraph within the graph to simplify its structure. The non-isolated nodes are renamed as subgraph pin names (e.g., SG1_VDD, SG1_VOUT, SG1_termA, etc) and added to the tokenizer table. AnalogGenie-Lite simplifies subgraphs only if the number of isolated nodes exceeds a certain threshold, balancing lightweight graph structures with the overhead in the tokenizer table. Figure 3 presents examples of AnalogGenie-Lite mined subgraphs with a large number of isolated nodes. Excitingly, these structures differ significantly from those used in previous work (Zhao & Zhang, 2022; 2020) based on human heuristics, underscoring the necessity of rigorous data mining.

### 3.3. Optimal Sequence Modeling

Harnessing an efficient data structure to model the circuit graph is the last key innovation of AnalogGenie-Lite's modeling method. Previous work (Dong et al., 2023; Lu et al., 2023) uses adjacency matrices to represent circuit graphs. However, an adjacency matrix requires $O\left(n^2\right)$ space to store $n$ nodes, regardless of the number of edges, which is inefficient for sparse graphs. Analog circuit topologies are typically sparse, as most devices connect only to their adjacent neighbors. In contrast, AnalogGenie-Lite optimally

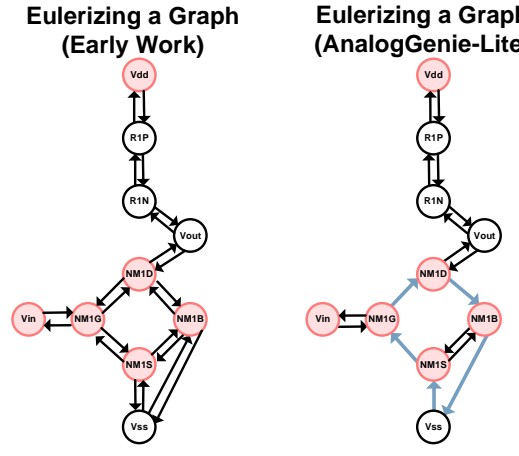

**Eulerizing a Graph (Early Work)**   **Eulerizing a Graph (AnalogGenie-Lite)**

**Early work Eulerian circuit  (23 tokens)**

['Vss' 'NM1S' 'NM1G' 'Vin' 'NM1G' 'NM1D' 'Vout' 'R1N' 'R1P' 'Vdd' 'R1P' 'R1N' 'Vout' 'NM1D' 'NM1B' 'Vss' 'NM1B' 'NM1S' 'NM1B' 'NM1D' 'NM1G' 'NM1S' 'Vss']

**AnalogGenie-Lite Eulerian circuit  (18 tokens)**

['Vss' 'NM1S' 'NM1G' 'Vin' 'NM1G' 'NM1D' 'Vout' 'R1N' 'R1P' 'Vdd' 'R1P' 'R1N' 'Vout' 'NM1D' 'NM1B' 'NM1S' 'NM1B' 'Vss']

*Figure 4.* Comparison between early work (Gao et al., 2025) and AnalogGenie-Lite Eulerizing method (red: odd degree nodes in the original undirected graph; blue: non-duplicated edges).

represents the circuit graph as the shortest closed path that visits every edge of a finite connected undirected graph at least once by solving the *Chinese Postman Problem*.

**Definition 3.1** (Chinese postman problem, (Edmonds &

Johnson, 1973)). Chinese postman problem is a combinatorial optimization problem that tries to find the shortest circuit that visits every edge of a finite connected undirected graph at least once.

When a graph contains an Eulerian circuit, that circuit represents the optimal solution.

**Definition 3.2** (Eulerian circuit, (Biggs et al., 1986)). Eulerian circuit is a graph trail that visits every edge exactly once and starts and ends at the same node.

However, not all finite connected undirected graphs contain Eulerian circuit unless all of their nodes have even degree.

**Definition 3.3** (Eulerian graph, (Biggs et al., 1986)). If all the nodes in a finite connected undirected graph have even degree, the graph is Eulerian and contains at least one Eulerian circuit.

Thus, the optimization problem for non-Eulerian graph is to find the smallest number of graph edges to duplicate so that the resulting multigraph does have an Eulerian circuit (Roberts & Tesman, 2024). As shown in Figure 4, existing non-optimal method (Gao et al., 2025) duplicates all the existing edges in a finite connected undirected graph to ensure the graph to be Eulerian by replacing each undirected edge $\{u, v\} \in E$ with two directed arcs $(u, v)$ and $(v, u)$ (i.e., traverse all the undirected edge exactly twice). This causes a large overhead in its sequence representation since not all the edges need to be duplicated to make the graph Eulerian. On the other hand, AnalogGenie-Lite finds the optimal sequence by employing Algorithm 1 (Mei-Ko, 1962; Edmonds & Johnson, 1973) to Eulerize the undirected graph. Since the topology graph $\mathcal{G} = (V, E)$ consists of unweighted edges and Algorithm1 requires a weighted undirected graph, we assign a weight of 1 to all edges in $\mathcal{G}$. The algorithm begins by identifying vertices with odd degrees. It then pairs the odd-degree vertices in a way that minimizes the total cost of connecting them, leveraging a shortest-path strategy. Finally, it duplicates the edges along these shortest paths to ensure all vertices have even degrees, thereby Eulerizing the graph. Through this process, AnalogGenie-Lite minimizes the duplication of existing edges, efficiently transforming the graph into an Eulerian circuit.

## 4. Results

### 4.1. Experiment Setup

**Datasets**: The AnalogGenie-Lite uses a dataset (Gao et al., 2025) that comprises 3350 unique and real-world topologies across 11 types: Op-Amps, LDOs, Bandgap references, Comparators, PLLs, LNAs, PAs, Mixers, VCOs, Power converters, and SC Samplers from public resources (Razavi, 2000; Razavi & Behzad, 2012; Johns & Martin, 2008; Gray et al., 2009; Allen & Holberg, 2011; Camenzind, 2005).

---

**Algorithm 1** Chinese Postman Algorithm

---

**Require:** A weighted, undirected graph $G = (V, E)$
1: **Step 1: Identify odd-degree vertices**
2: odd_vertices $\leftarrow \{v \in V \mid \text{degree}(v) \mod 2 \neq 0\}$
3: **Step 2: Pair up odd-degree vertices to minimize cost**
4: CompleteGraph $\leftarrow$ Construct a complete graph where vertices are the odd-degree vertices, and edge weights are the shortest path distances between each pair of odd-degree vertices in the original graph.
5: pairs $\leftarrow$ Solve the minimum weight perfect matching problem on CompleteGraph using the Blossom algorithm (Edmonds, 1965).
6: **Step 3: Duplicate edges to make degrees even**
7: **for** each pair $(u, v) \in$ pairs **do**
8:     path $\leftarrow$ ShortestPath(u, v, G) using Dijkstra algorithm (Dijkstra, 2022)
9:     **for** each edge $(x, y) \in$ path **do**
10:         Add duplicate edge $(x, y)$ to $G$
11:     **end for**
12: **end for**

---

**Training setup**: During pretraining, AnalogGenie-Lite split the data into train and validation sets with a 9 to 1 ratio. It augments the dataset by generating multiple unique Eulerian circuits per topology. It uses a decoder-only transformer with 6 layers, 6 attention heads, and 11.825 million parameters, with a vocabulary size of 1029 and a maximum sequence length of 1024. For performance evaluation, AnalogGenie-Lite leverages reinforcement learning with human feedback (Ouyang et al., 2022) to target specific types of analog circuits optimized for given performance metrics.

**Baseline**: We select AnalogCoder (Lai et al., 2024) and Artisan (Chen et al., 2024) as the representative of text generation work. Then, we select CktGNN (Dong et al., 2023), LaMAGIC (Chang et al., 2024), and AnalogGenie (Gao et al., 2025) as the representative of graph generation work. The differences between these methods and AnalogGenie-Lite are discussed in Section 2. We follow original work to produce their results.

**Evaluation tasks and metrics**: We evaluate the generative quality of each method using the following metrics:
(1) Validity: An unsized circuit is considered valid if it can be simulated in SPICE without errors (e.g., floating or shorted nodes). Each method generates 1,000 topologies, and we report the percentage of valid designs.
(2) Scalability: The largest valid circuit generated by each model, based on the number of devices.
(3) Versatility: The versatility of a model is determined by the number of distinct analog circuit types it generates.
(4) Novelty: To evaluate novelty, each method generates 1,000 topologies, and we measure the percentage of these that differ from those in the dataset. Topology differences

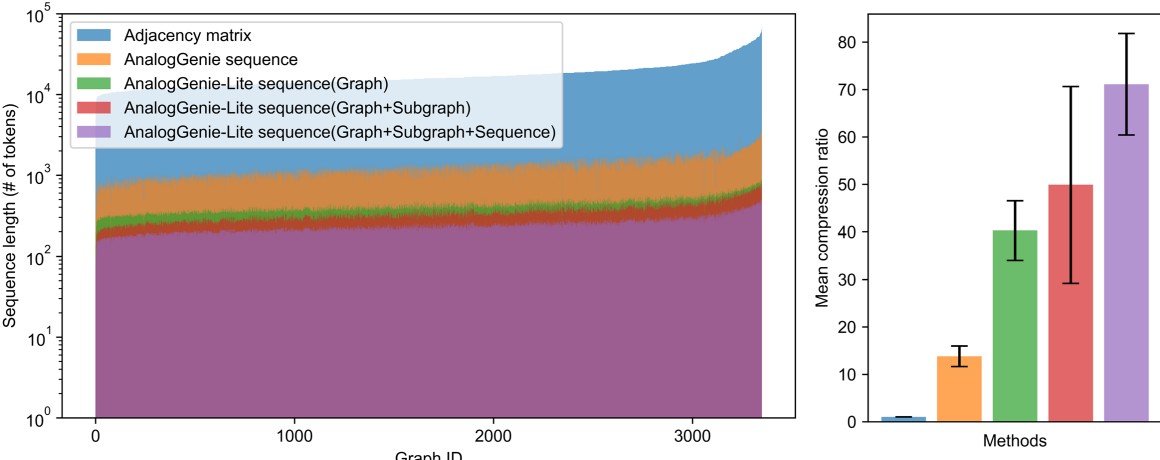

*Figure 5.* Comparison between adjacency matrix (Dong et al., 2023; Lu et al., 2023), AnanlogGenie (Gao et al., 2025) sequence, and AnanlogGenie-Lite sequence for representing each topology or graph within the dataset in terms of sequence length (left) and mean compression ratio (right). We flatten adjacency matrices to sequences for comparison.

are quantified by converting the circuits into graphs and computing the maximum mean discrepancy (MMD) (Guo & Zhao, 2022) between these generated graphs and real-world graphs derived from the dataset.

(5) Performance: Each generated topology is sized using a genetic algorithm, and the resulting figure-of-merit (FoM) – considering all major metrics (e.g., gain, bandwidth, and power for operational amplifiers) – is used as a comprehensive performance indicator. We compare the best FoM achieved by circuits generated from each model.

### 4.2. Evaluating the Compression Rate

We begin by evaluating AnalogGenie-Lite's lightweight graph modeling in terms of mean compression ratio for representing circuit topology. Figure 5 compares the sequence length required to encode each topology using AnalogGenie-Lite, the adjacency matrix baseline (Dong et al., 2023; Lu et al., 2023), and the state-of-the-art sequence representation, AnalogGenie (Gao et al., 2025). The adjacency matrix approach, based on the device-pin graph (Gao et al., 2025), results in sequences of approximately $10^4$ to $10^5$ tokens due to its $O(n^2)$ space complexity, making it impractical for training with typical LLMs (Radford et al., 2019; Brown et al., 2020; Achiam et al., 2023). AnalogGenie mitigates this issue by encoding graphs as non-optimal Eulerian circuits, reducing sequence length by $13.81\times$ to around $10^3$ tokens. However, it still struggles with large circuits exceeding 100 devices, where sequences reach $10^4$ tokens. To enhance scalability, AnalogGenie-Lite employs a more efficient graph modeling strategy. The AnalogGenie-Lite (Graph) method prunes redundant nodes and edges while maintaining the non-optimal Eulerian representation, improving compression by $40.28\times$ over adjacency matrices and

$2.91\times$ over AnalogGenie. The AnalogGenie-Lite sequence (Graph+Subgraph) further prunes isolated nodes and edges within subgraphs, improving compression to $49.91\times$ and $3.61\times$ over the respective baselines. However, the compression benefit varies across circuits as shown in Figure 5, particularly for those with unique structures (e.g., PA circuits), emphasizing the need for a flexible generation approach capable of predicting both device pins and subcircuits. Finally, AnalogGenie-Lite (Graph+Subgraph+Sequence) integrates all optimizations, solving the *Chinese Postman Problem* for an optimal Eulerian circuit representation. This method achieves a $71.11\times$ compression over adjacency matrices and $5.15\times$ over AnalogGenie. These results demonstrate that AnalogGenie-Lite offers a scalable and precise representation, crucial for robust circuit topology discovery.

### 4.3. Evaluating the Generation Quality

Then, we evaluate AanalogGenie-Lite's generation quality by comparing to prior methods, as shown in Table 1:

**Validity:** AnalogGenie-Lite achieves 97% validity, significantly surpassing prior methods. AnalogCoder generates only 63.4% valid circuits due to errors in code generation, while CktGNN, LaMAGIC, and AnalogGenie achieve 66.3%, 72%, and 73.5% validity, respectively, using graph-based techniques. Artisan, focused on topology selection, reaches 82% validity but cannot generate novel circuits. In contrast, AnalogGenie-Lite's lightweight graph modeling mitigates error propagation, achieving a 23.5% to 33.6% improvement over previous methods while generating over 99% novel circuits.

**Scalability:** AnalogGenie-Lite achieves unmatched scalability. CktGNN and LaMAGIC, limited by quadratic costs

*Table 1.* Performance comparison between AnalogGenie-Lite and existing analog circuit topology generation work.

| Evaluation metric | Validity (%) ↑ | Scalability ↑ | Versatility ↑ | Novelty | | FoM | |
| --- | --- | --- | --- | --- | --- | --- | --- |
| | | | | Diff circuit (%) ↑ | MMD ↓ | Op-Amp ↑ | Power converter ↑ |
| AnalogCoder (Lai et al., 2024) | 63.4 | 10 | 7 | 0 | 0 | 233.4 | N/A |
| AnalogGenie (Gao et al., 2025) | 73.5 | 63 | 11 | 98.8 | 0.0532 | 13744.8 | 3.3 |
| Artisan (Chen et al., 2024) | 82 | 18 | 1 | 0 | 0 | 12769.5 | N/A |
| CktGNN (Dong et al., 2023) | 66.3 | 22 | 1 | 93 | 0.313 | 943.2 | N/A |
| LaMAGIC(Chang et al., 2024) | 72 | 4 | 1 | 3 | 0.001 | N/A | 2.7 |
| **AnalogGenie-Lite (Graph)** | **80.1** | **183** | **11** | **99.7** | **0.0498** | **14727.3** | **3.71** |
| **AnalogGenie-Lite (Graph+Subgraph)** | **87.3** | **227** | **11** | **99.8** | **0.0467** | **14767.5** | **3.76** |
| **AnalogGenie-Lite (Graph+Subgraph+Sequence)** | **97** | **324** | **11** | **99.8** | **0.0408** | **15017.7** | **4.02** |

of adjacency matrix representations, handle circuits with up to 4 and 22 devices, respectively. AnalogCoder and Artisan, constrained by prompt templates, generate circuits with a maximum of 10 and 18 devices. AnalogGenie improves scalability to 63 devices with an efficient sequence representation. AnalogGenie-Lite, leveraging a lightweight representation, extends this further to 324 devices, achieving a $5.15\times$ to $71.11\times$ improvement over prior methods.

**Versatility:** AnalogGenie-Lite surpasses Artisan, CktGNN, and LaMAGIC, which are restricted to designing only a single circuit type. AnalogCoder, while supporting seven types, is limited by a synthesis library containing 20 topologies.

**Novelty:** AnalogGenie-Lite outperforms other methods in discovering novel circuits. AnalogCoder and Artisan primarily reuse existing topologies or subblocks, while LaMAGIC is restricted to a small design space, significantly limiting its ability to explore new topologies. Although both Ckt-GNN and AnalogGenie-Lite support the generation of larger circuits, CktGNN is trained on synthetic datasets that lack critical real-world features. AnalogGenie-Lite, trained on real-world circuit, improves the MMD metric by over $7.67\times$ and generates approximately 99% novel circuits. Compared to AnalogGenie, AnalogGenie-Lite produces more realistic circuits with lower MMD values due to its lightweight graph modeling, which eliminates redundant edges and nodes and lets the model focus on generating essential components.

**Performance:** In op-amp design, AnalogCoder achieves a low FoM of 233.4 due to limited design options, while Ckt-GNN performs better at 943.2 by selecting optimized subcircuits. Artisan achieves an even higher FoM of 12769.5 by selecting state-of-the-art designs. AnalogGenie and AnalogGenie-Lite by pretraining on diverse circuit topologies, discover unseen topologies with superior FoM values of 13744.8 and 15017.7, respectively, outperforming dedicated synthesizers. AnalogGenie-Lite's advantage comes from its lightweight graph representation, eliminating redundant circuit details while focusing on learning essential components. A similar trend is seen in power converter design, where AnalogGenie-Lite achieves an FoM of 4.02, surpassing AnalogGenie (3.3) and LaMAGIC (2.7).

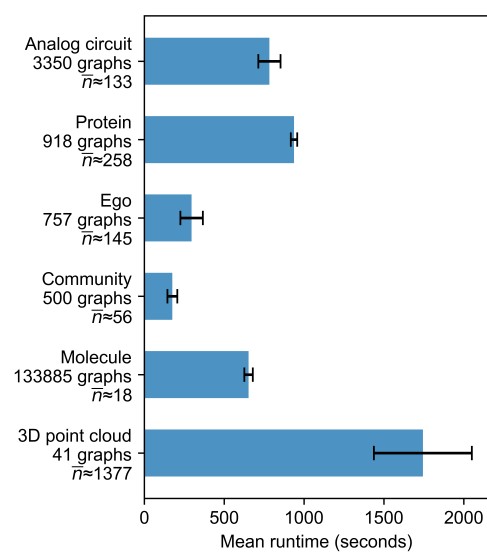

*Figure 6.* Runtime of Algorithm 1 on processing circuit (Gao et al., 2025), protein (Dobson & Doig, 2003), ego (Sen et al., 2008), community (ERD & Renyi, 1959), molecule (Ramakrishnan et al., 2014), and 3D point cloud (Neumann et al., 2013) graph datasets.

### 4.4. Evaluating the Computation Cost

We begin by conducting a detailed theoretical analysis of Algorithm 1's time complexity for solving the Chinese postman problem (CPP). As noted in (Grötschel & Yuan, 2012), our approach reduces CPP to a matching problem, which can be solved in polynomial time. In particular, Theorem 12.10 in Section 12.2 (page 307) of (Korte et al., 2011) shows that the minimum weight perfect matching problem can be solved in $O\left(n^3\right)$, where $n$ is the number of nodes.

Next, we empirically evaluate the computational cost of Algorithm 1. Our experiments were conducted on a PC equipped with a 16-core i9-12900KF 3.20 GHz CPU and 32 GB of RAM. Figure 6 shows the mean runtime results for a total of 5 runs on diverse graph datasets. Our Python program for Algorithm 1 requires at most 29 minutes and 4.36 seconds on average to preprocess the entire dataset. Importantly, this preprocessing step is performed only once during

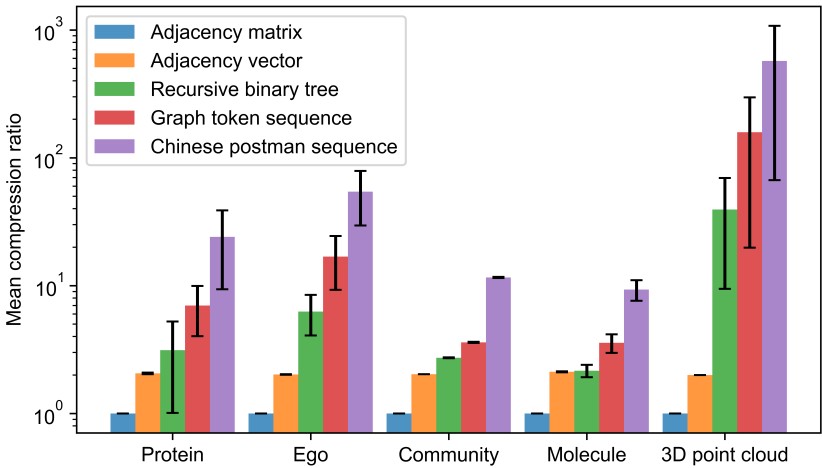

*Figure 7.* Comparison between adjacency matrix, adjacency vector (You et al., 2018b), recursive binary tree (Dai et al., 2020), graph token sequence (Chen et al., 2025) and our Chinese postman sequence for protein (Dobson & Doig, 2003), ego (Sen et al., 2008), community (ERD & Renyi, 1959), molecule (Ramakrishnan et al., 2014), and 3D point cloud (Neumann et al., 2013) graphs.

training. This one-time cost is justified by the substantial benefits in compression and scalability that our approach offers.

### 4.5. Cross-Domain Case Study

Finally, beyond circuits, we believe that AnalogGenie's lightweight graph modeling can also be applied to other domains, such as protein generation (Jumper et al., 2021), personalized recommendation (Epasto et al., 2015), community detection (Fortunato, 2010), drug discovery (You et al., 2018a), and 3D object recognition (Shi & Rajkumar, 2020) that can also be represented as graphs. To demonstrate its general applicability, we conduct a case study on protein (Dobson & Doig, 2003), ego (Sen et al., 2008), community (ERD & Renyi, 1959), molecule (Ramakrishnan et al., 2014), and 3D point cloud (Neumann et al., 2013) graphs. Besides the basic adjacency matrix, we evaluate AnalogGenie-Lite's Chinese postman sequence by comparing it with the following graph data structures that have been developed in recent years.

**Adjacency vector** (You et al., 2018b): For a given node ordering $\pi$, each node $\pi(v_i)$ (except the first) is encoded via an adjacency vector $S_i^\pi$ that captures its connections to all previously added nodes. Without ordering constraints, the length of these vectors grows linearly (e.g. the first node has no vector, the second one entry, the third two entries, etc.), resulting in an overall space complexity of $O(n^2)$.

**Recursive binary tree** (Dai et al., 2020): This method uses a binary tree-structured conditioning on the matrix's rows and columns to reduce the complexity from $O(n^2)$ to $O((n+m)\log n)$, where $n$ is the number of nodes and $m$

the number of edges.

**Graph token sequence** (Chen et al., 2025): In this approach, the sequence consists of two parts. The first part is a tuple to represent all the nodes' definitions in the graph (e.g., its type and unique index), followed by a special token marking the transition to the edge definition. In the second part, each edge is then encoded as a triple containing the source node index, the destination node index, and the type. The space complexity of it is $O(n+m)$.

As illustrated in Figure 7, AnalogGenie-Lite's Chinese Postman sequence achieves compression ratios of $9.33\times$ to $572.56\times$ over adjacency matrices and $2.61\times$ to $3.61\times$ over the strongest prior-art approach (graph token sequences). By definition, the Chinese Postman sequence is the shortest closed path that traverses every edge of an undirected graph at least once, resulting in a linear space complexity of $O(m)$. These results confirm that sparse graphs—3D point clouds, ego networks, and protein interaction graphs—satisfy $m << n^2$ and compress exceptionally well, while denser graphs (community and molecular) with $m$ closer to $n^2$ remain more challenging.

## 5. Conclusion and Future Work

This work presents AnalogGenie-Lite, a decoder-only transformer model for discovering novel analog IC topologies with enhanced scalability and precision through lightweight graph modeling. While it sets a new standard in circuit topology discovery, several challenges remain. Key issues include efficiently mining representative subgraphs from diverse datasets and further improving the compression while preserving essential features for effective model learning.

## Acknowledgements

This work was partially supported by NSF Award #2526432.

## Impact Statement

"This paper presents work whose goal is to advance the field of Machine Learning. There are many potential societal consequences of our work, none which we feel must be specifically highlighted here."

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

## A. Appendix

### A.1. More Details about Redundant Edges Pruning

As shown in Figure 8a, a total of seven pins—PM1S, PM1B, PM2S, PM2B, PM3S, PM3B, and Vdd—share the same net within the topology. The device-pin graph (Gao et al., 2025) connects each pair of these pins directly, forming a fully connected subgraph that requires $\frac{7\times(7-1)}{2} = 21$ edges to represent the shared net. In contrast, AnalogGenie-Lite selects Vdd and connects the remaining pins directly to it. This avoids overlapping device-pin cycles, effectively pruning redundant edges and reducing the edge count to just $7 - 1 = 6$, achieving a $3.5\times$ reduction in edges compared to the device-pin graph.

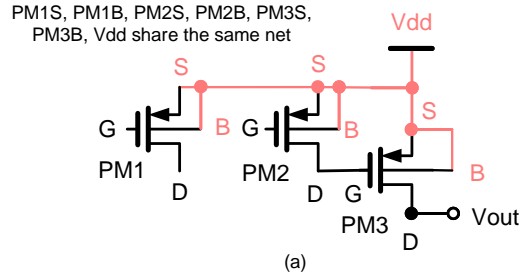

**Circuit Topology**

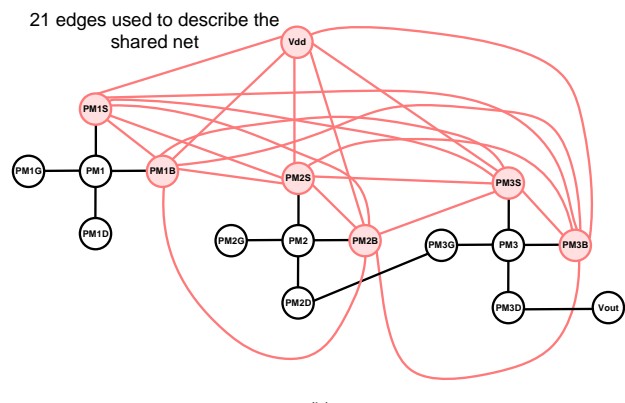

**Device Pin Graph**

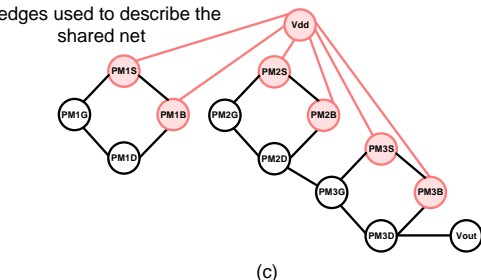

**AnalogGenie-Lite Graph**

*Figure 8.* An example of circuit topology, device-pin graph (Gao et al., 2025), and AnalogGenie-Lite graph.

## A.2. Tokenizer Lookup Table

Table 2 presents the tokenizer table used by AnalogGenie-Lite for its versatile generation scheme, which predicts the next device, subcircuit, or circuit pin. The token indices are structured as follows: indices 0 to 549 represent basic device pins, covering NMOS, PMOS, NPN, PNP, resistors, capacitors, inductors, and diodes. Indices 550 to 890 correspond to subcircuit pins, including mined subgraphs, XOR gates, inverters, transmission gates, etc. Lastly, indices 891 to 1027 define circuit pins such as VIN, IIN, VTRACK, VDD, VSS, etc. The TRUNCATE token (index 1028) is used to pad sequences to a uniform length for training.

*Table 2.* Tokenizer look-up table with basic devices, subcircuits, and circuit pins.

| Device | Index | Device | Index | Device | Index |
|--------|-------|--------|-------|--------|-------|
| NM1D | 0 | NM1G | 1 | NM1S | 2 |
| NM1B | 3 | NM2D | 4 | ... | ... |
| NM25B | 99 | PM1D | 100 | PM1G | 101 |
| PM1S | 102 | PM1B | 103 | PM2D | 104 |
| ... | ... | PM25B | 199 | NPN1C | 200 |
| NPN1B | 201 | NPN1E | 202 | NPN2C | 203 |
| ... | ... | NPN25E | 274 | PNP1C | 275 |
| PNP1B | 276 | PNP1E | 277 | PNP2C | 278 |
| ... | ... | PNP25E | 349 | R1P | 350 |
| R1N | 351 | R2P | 352 | ... | ... |
| R25N | 399 | C1P | 400 | C1N | 401 |
| C2P | 402 | ... | ... | C25N | 449 |
| L1P | 450 | L1N | 451 | L2P | 452 |
| ... | ... | L25N | 499 | DIO1P | 500 |
| DIO1N | 501 | DIO2P | 502 | ... | ... |
| DIO25N | 549 | SG1VDD | 550 | SG1VOUT | 551 |
| SG1TA | 552 | SG1TB | 553 | SG1TC | 554 |
| SG1TD | 555 | SG1TE | 556 | SG2VDD | 557 |
| ... | ... | XOR1A | 726 | XOR1B | 727 |
| XOR1VDD | 728 | XOR1VSS | 729 | XOR1Y | 730 |
| ... | ... | XOR5Y | 750 | INV1A | 751 |
| INV1Q | 752 | INV1VDD | 753 | INV1VSS | 754 |
| INV2A | 755 | ... | ... | INV10VSS | 790 |
| TG1A | 791 | TG1B | 792 | TG1C | 793 |
| TG1VDD | 794 | TG1VSS | 795 | TG2A | 796 |
| ... | ... | TG20VSS | 890 | VIN1 | 891 |
| VIN2 | 892 | VIN3 | 893 | VIN4 | 894 |
| VIN5 | 895 | IIN1 | 896 | IIN2 | 897 |
| ... | ... | VTRACK1 | 1024 | VTRACK2 | 1025 |
| VDD | 1026 | VSS | 1027 | TRUNCATE | 1028 |

