# OpenReview forum: "AnalogGenie-Lite: Enhancing Scalability and Precision in Circuit Topology Discovery through Lightweight Graph Modeling"
_ICML.cc/2025/Conference — ICML 2025 poster_

### Official Review · Reviewer_Xhyg · 2025-03-13

**Overall Recommendation:** 4

**Summary:**

This work introduces a decoder-only transformer for analog topology generation by solving three critical challenges. At the graph level, it simplifies current approach by removing redundant nodes and edges. At the sub-graph level, it employs subsequent subgraph minging to identify commonly reused subgraphs in the database.  Lastly, it models a circuit graph as a shortest closed path by solving the Chinese Postan problem.

**Claims And Evidence:**

The three techniques in the paper are well supported by the experimental results.

**Essential References Not Discussed:**

No.

**Experimental Designs Or Analyses:**

From the formulation of the task, it is possible to leverage pre-trained language model, e.g., old models such as T5 or new models such as Llama and GPT-4o, as a prior and conduct fine-tuning on it. It would be better if there are discussion about them and a comparison with current train-from-scratch approach.

**Methods And Evaluation Criteria:**

This paper introduces though evaluation metrics during the experiments. It would be better if there is a discussion for time cost of each method. The two techniques introduced in this work, subgraph mining and solving Chinese Postman problem, seems to introduce much time.

**Other Comments Or Suggestions:**

No additional comments.

**Other Strengths And Weaknesses:**

No additional comments.

**Questions For Authors:**

1) the generation task in this work seems to mean unconditional generation, where we cannot control what will be generated. Is that right? Can it be used for text-based generation, where text describes the design requirements.

**Relation To Broader Scientific Literature:**

It proposes a compact way to represent circuit topology. Although it is only trained on unconditional generation task, it can be used on conditional generation (e.g., text as condition) in the future.

**Theoretical Claims:**

There is no theoretical claim.

---

> ### Author Rebuttal · Authors · 2025-03-31
>
> We appreciate the reviewer's valuable comments.
>
> > **Q1:**  It would be better if there is a discussion for time cost of each method. The two techniques introduced in this work, subgraph mining and solving Chinese Postman problem, seems to introduce much time.
>
> We would like to evaluate the time cost of our subgraph mining algorithm and Algorithm 1 for solving the Chinese postman problem empirically. Our experiment is conducted on a PC with a 16‐core i9-12900KF CPU and 32 GB RAM. We run each algorithm 5 times on the analog circuit dataset. The mean runtime for the subgraph mining algorithm is 1473.74 seconds, and Algorithm 1 is 783.30 seconds. **This one-time preprocessing cost can be amortized by the substantial memory and time savings during model pre-training**.
>
> > **Q2:**  From the formulation of the task, it is possible to leverage pre-trained language model, e.g., old models such as T5 or new models such as Llama and GPT-4o, as a prior and conduct fine-tuning on it. It would be better if there are discussion about them and a comparison with current train-from-scratch approach.
>
> We appreciate the reviewer’s suggestion. In our study, we have already compared our train-from-scratch approach with methods that leverage pre-trained language models. Specifically, our experiments included:
>
> - AnalogCoder [1]: a **GPT-4o-based model** using domain-specific prompt engineering.
> - LaMAGIC [2]: a model fine-tuned on a **FLAN-T5-base** backbone.
> - Artisan [3]: a model fine-tuned on a **Llama2-7b** backbone.
>
> As detailed in Table 1 of our paper, our approach outperforms these baselines across several key metrics, including validity, scalability, versatility, novelty, and FoM.
>
> > **Q3:**  the generation task in this work seems to mean unconditional generation, where we cannot control what will be generated. Is that right? Can it be used for text-based generation, where text describes the design requirements.
>
> **Our approach is not limited to unconditional generation**. We incorporate mechanisms to **steer the generation process toward a specific type of analog circuit topologies optimized for key performance metrics** (e.g., Figure of Merit) by leveraging **reinforcement learning with human feedback (RLHF)** to fine-tune our pre-trained model. The process begins with training a reward model that evaluates generated topologies based on validity, circuit type, and performance using a dataset labeled by humans. Once this reward model is trained, we apply proximal policy optimization (PPO)  [4] to iteratively refine the pre-trained model. In each training epoch, the model generates a batch of candidate topologies that are scored by the reward model, and the model parameters are adjusted to maximize the expected accumulated reward scores, effectively steering the generation toward high-performance designs. **Although AnalogGenie-Lite currently does not support text-based input, its underlying next-token-prediction mechanism makes it naturally compatible with text generation**, and by augmenting the netlist dataset with text-based descriptions of design requirements, we can extend our approach to **include text-based control in future iterations (as acknowledged by the reviewer in Relation To Broader Scientific Literature section)**.
>
>
>
> [1] Lai, Yao, et al. "Analogcoder: Analog circuit design via training-free code generation." *arXiv preprint arXiv:2405.14918* (2024).
>
> [2] Chang, Chen-Chia, et al. "Lamagic: Language-model-based topology generation for analog integrated circuits." *arXiv preprint arXiv:2407.18269* (2024).
>
> [3] Chen, Zihao, et al. "Artisan: Automated operational amplifier design via domain-specific large language model." *Proceedings of the 61st ACM/IEEE Design Automation Conference*. 2024.
>
> [4] Ouyang, Long, et al. "Training language models to follow instructions with human feedback." *Advances in neural information processing systems* 35 (2022): 27730-27744.

---

### Official Review · Reviewer_EghR · 2025-03-14

**Overall Recommendation:** 3

**Summary:**

AnalogGenie-Lite presents a decoder-only framework designed to discover novel analog circuit topologies by leveraging lightweight graph modeling. Its key contributions lie in three innovations:
- A precise and efficient graph modeling approach that prunes redundant nodes and edges.
- Optimal sequence modeling via solving the Chinese Postman Problem to derive near-optimal Eulerian circuits.
Experimental results on a dataset of over 3350 real-world topologies across 11 analog circuit types demonstrate marked improvements.

**Claims And Evidence:**

**Main Claims**:
- AnalogGenie-Lite achieves a dramatic reduction in sequence length (up to 71.11× over traditional adjacency matrix representations) and substantial improvements in validity.
- The method is broadly applicable, as evidenced by case studies extending to domains like protein graphs and social networks.

**Evidence Provided**:
- Quantitative comparisons (e.g., Table 1 and Figure 5) show improved compression ratios, higher validity percentages, and enhanced performance metrics (FoM) over baselines.
- Algorithmic pseudocode support the claims regarding efficiency and optimality in sequence .

**Essential References Not Discussed:**

N/A

**Experimental Designs Or Analyses:**

**Design**:
- 3350 unique analog circuit topologies spanning 11 types.
- The experimental design is robust, with ablation studies to isolate the contribution of each innovation (pruning, subgraph mining, and optimal sequencing).

**Methods And Evaluation Criteria:**

**Methods**:
- Lightweight Graph Modeling and Optimal Sequence Modeling

**Evaluation Criteria**:
- Validity: Percentage of generated circuits that are SPICE-simulatable without errors.
- Scalability: Maximum circuit size (in terms of devices) that can be generated.
- Versatility & Novelty: Diversity of analog circuit types generated and the percentage of designs that differ from those in the training dataset.
- Performance (FoM): A figure-of-merit combining key circuit performance metrics such as gain, bandwidth, and power.

**Other Comments Or Suggestions:**

More detailed analysis regarding the integration of reinforcement learning with human feedback—specifically, how this impacts circuit FoM—could further enhance the paper.

**Other Strengths And Weaknesses:**

**Strengths**:
- Innovation: The paper introduces a highly innovative and technically rigorous approach to analog circuit topology generation.
- Efficiency: The significant reduction in sequence length and improved compression ratios are compelling, addressing a critical bottleneck in training large language models on graph representations.

**Weaknesses**:
- Ablation: Additional discussion on the sensitivity of performance to hyperparameter choices in the pruning and subgraph mining processes could be beneficial.

**Questions For Authors:**

1.	How sensitive is the overall performance to the threshold used for pruning isolated nodes in subgraph modeling?
2.	Could you elaborate on how reinforcement learning with human feedback is integrated into the generation process, in detail?
3.	How does AnalogGenie-Lite handle circuits with highly irregular or non-repetitive topologies that may not benefit from subgraph simplification?

**Relation To Broader Scientific Literature:**

N/A

**Theoretical Claims:**

N/A

---

> ### Author Rebuttal · Authors · 2025-03-31
>
> Thanks for the reviewer's valuable and constructive feedback.
>
> > **W1 & Q1:** Ablation: Additional discussion on the sensitivity of performance to hyperparameter choices in the pruning and subgraph mining processes could be beneficial. How sensitive is the overall performance to the threshold used for pruning isolated nodes in subgraph modeling?
>
> We conducted an ablation study to assess how AnalogGenie-Lite's performance responds to two key hyperparameters in our subgraph pruning process: (1) the frequency threshold of subgraph occurrence in the dataset and (2) the threshold for the number of isolated nodes in a subgraph. Specifically, we experimented with frequency thresholds of 5%, 25%, and 50% along with isolated node thresholds of 4 and 8. A subgraph is pruned if its occurrence frequency exceeds the chosen threshold and if it contains more isolated nodes than the specified limit. After compressing the dataset, we followed our standard pretraining and finetuning procedure for AnalogGenie-Lite. The results are summarized in the table below:
>
> |                       | Validity (%) $\uparrow$ | Scalability $\uparrow$ | MMD $\downarrow$ | Op-Amp FoM $\uparrow$ | Power converter FoM $\uparrow$ |
> | --------------------- | ----------------------- | ---------------------- | ---------------- | --------------------- | ------------------------------ |
> | 5% freq & 4 isolated  | 96.9                    | 341                    | 0.0419           | 15015.9               | 3.99                           |
> | 5% freq & 8 isolated  | 97.1                    | 330                    | 0.0409           | 15016.4               | 4.00                           |
> | 25% freq & 4 isolated | 97.3                    | 324                    | 0.0408           | 15017.7               | 4.02                           |
> | 25% freq & 8 isolated | 96.5                    | 322                    | 0.0409           | 15012.5               | 4.01                           |
> | 50% freq & 4 isolated | 95.3                    | 306                    | 0.0420           | 14998.3               | 3.99                           |
> | 50% freq & 8 isolated | 94.2                    | 293                    | 0.0433           | 14982.8               | 3.99                           |
>
> Overall, **most performance metrics are relatively insensitive to these hyperparameters**. The one exception is scalability, which is notably affected by changes in the thresholds due to their direct impact on the number of subgraph pruning candidates. Notably, in 5% frequency results, aggressively lowering the thresholds to achieve higher compression introduces a large number of infrequently used special tokens into the tokenizer, which adversely affects training and ultimately hurts performance on metrics other than scalability.
>
> > **C1 & Q2:** More detailed analysis regarding the integration of reinforcement learning with human feedback—specifically, how this impacts circuit FoM—could further enhance the paper. Could you elaborate on how reinforcement learning with human feedback is integrated into the generation process, in detail?
>
> AnalogGenie-Lite pretrained-only model can initially produce a wide variety of analog circuit topologies without inherent bias toward specific performance metrics. To optimize it for generating high-performance circuits, we integrate RLHF in a two-step fine-tuning process. First, **a reward model—trained on human-labeled examples—is used to score generated circuit topologies based on type and performance**.  Then, we **fine-tune the pretrained model with PPO: the model generates new designs, the reward model assigns scores, and PPO updates the model to maximize the expected accumulated reward**. This process directs the pretrained model toward optimal circuits, boosting the FoM for Op-Amps from 291.3 to 15017.7 and for power converters from 2.6 to 4.02.
>
> > **Q3:** How does AnalogGenie-Lite handle circuits with highly irregular or non-repetitive topologies that may not benefit from subgraph simplification?
>
> Our approach is robust even for highly irregular circuit topologies that do not benefit from subgraph simplification. Specifically, we **exploit a common feature in all analog circuits: the ground node, which connects to multiple device pins within the circuit**. By consolidating these multi-pin shared edge connections through graph-level simplification, AnalogGenie-Lite can maintain compression rate without subgraph simplification. As shown in Figure 5, **our graph-level method delivers a mean compression rate of 40.28$\times$ compared to adjacency matrix representation without the subgraph simplification**. Furthermore, because **analog circuit graphs are inherently sparse**, our optimal sequence modeling strategy can **maintain an additional 1.43$\times$ improvement in compression performance**. These two approaches ensure that our method remains effective even when subgraph simplification is less applicable.

---

### Official Review · Reviewer_4Q4T · 2025-03-14

**Overall Recommendation:** 3

**Summary:**

The paper proposes AnalogGenie-Lite, a generative model for discovering analog circuit topologies using lightweight graph modeling. The main contributions of this work are converting device-pin representations from graph to sequence and employing LLMs to design the compressed sequence which implicitly contains topological information. In experiments, this work validates the effectiveness on circuit discover and does case study on protein and social analysis.

**Claims And Evidence:**

Yes. The claims are generally convincing.

**Essential References Not Discussed:**

None.

**Experimental Designs Or Analyses:**

Why the baseline, AnalogGenie, is the version without fine-tuning? This work also does fine-tuning by reinforcement learning with human feedback. It seems that there is an unfair comparison.

**Methods And Evaluation Criteria:**

Yes. The benchmarks look reasonable.

**Other Comments Or Suggestions:**

None.

**Other Strengths And Weaknesses:**

Strengths:
1. The discussed problem is interesting and important.
2. The results are generally promising and good.


Weaknesses:
1. While the authors compress the graph to sequence, which reduce the complexity of the problem to linear, it should also be important analyze the computational complexity of the 'Chinese postman problem'.
2. Could the authors also discuss other algorithms converting graph to sequence? I am just curious why the authors choose this method to compress the graph.
3. Graphs can also be compressed by other algorithms, the simplest one is just record the edges and their corresponding nodes. However,  the authors only compare it with vanilla adjacency matrix and sounds like overclaim their scalability (compression rate).
4.  Why only the authors only conduct case study for protein and ego graphs? While discovering circuit topologies is an interesting problem, I think the contribution will be greatly improved if the proposed methods can be applied to more fields.

**Questions For Authors:**

Please answer my question in experiments setting and weaknesses. I am happy to turn to positive scores if the answers are satisfactory.

**Relation To Broader Scientific Literature:**

This work employs a new way to represent the graph and uses LLMs to model it.

**Theoretical Claims:**

The authors do not prove the claims theoretically.

---

> ### Author Rebuttal · Authors · 2025-03-31
>
> Thanks for the constructive comments. We address your concerns below.
>
> > **Q1:** Why the baseline, AnalogGenie, is the version without fine-tuning? This work also does fine-tuning by reinforcement learning with human feedback. It seems that there is an unfair comparison.
>
> Our evaluation is designed to ensure **fairness between our work and the baseline AnalogGenie**. As discussed in Section 4.1 on the training setup, **our work is not fine-tuned for evaluations of validity, scalability, versatility, and novelty. We apply RLHF only when evaluating its performance on a specific circuit type**, thereby optimizing it for high-performance circuits. This approach allows us to assess our work both as a versatile foundation model and as a specialized agent. For our baseline, AnalogGenie is evaluated under the same conditions—it is **pre-trained solely for assessing validity, scalability, versatility, and novelty. For performance evaluation, AnalogGenie is also fine-tuned with RLHF**.
>
> > **Q2:** While the authors compress the graph to sequence, which reduce the complexity of the problem to linear, it should also be important analyze the computational complexity of the 'Chinese postman problem'.
>
> We would like to empirically evaluate the computational cost of Algorithm 1 for solving the Chinese postman problem. Our experiments on a 16‐core i9-12900KF (32 GB RAM) show that Algorithm 1 takes, on average, 937.36 seconds for the Protein dataset while the Analog circuit and Ego datasets took 783.30 and 296.06 seconds, respectively. **This one-time preprocessing cost is justified by the substantial memory savings during training**.
>
> > **Q3:** Could the authors also discuss other algorithms converting graph to sequence? I am just curious why the authors choose this method to compress the graph.
>
> Prior method like **adjacency vector** encodes each node by its connections to all preceding nodes, which results in a $O\left(n^2\right)$ complexity ($n$ is the number of nodes and $m$ is the number of edges). Alternatives include a **recursive binary tree [3]** that conditions on matrix rows and columns, reducing complexity to $O((n+m) \log n)$, and a **graph token sequence[4]** that first encodes node definitions followed by edge, achieving $O(n+m)$.  We choose the Chinese postman sequence for two reasons. First, it has a complexity of $O\left(m\right)$, which is **efficient for representing sparse graphs like analog circuits**. Second, representing the graph as a traversal path not only **preserves critical structure information** but also **aligns well with the behavior of language models** by predicting the next token based on the current context.
>
> > **Q4:** Graphs can also be compressed by other algorithms, the simplest one is just record the edges and their corresponding nodes. However, the authors only compare it with vanilla adjacency matrix and sounds like overclaim their scalability (compression rate).
>
> We appreciate the reviewer's suggestion. While an edge list is compact, **prior work LaMAGIC has already explored this approach by representing graphs as a list of hyperedges**. In that work, the **edge-list underperformed compared to the adjacency matrix**. This indicates that, despite its compactness, the oversimplified structure of an edge list makes it harder for language models to learn and generate complex graph patterns. In contrast, our method is designed to **not only compress the graph but also to preserve structural information critical for effective generation**. Moreover, **our experiments are not limited to comparisons with the adjacency matrix**. We also compare our work with the **AnalogGenie sequence in Figure 5** and the **adjacency vector in Figure 6**. In response to the reviewer's valuable suggestion, **we will extend our case study in Q5 to include additional graph compression algorithms**.
>
> > **Q5:** Why only the authors only conduct case study for protein and ego graphs? While discovering circuit topologies is an interesting problem, I think the contribution will be greatly improved if the proposed methods can be applied to more fields.
>
> We have expanded our case study to include molecules [1] and 3D point cloud graphs [2]. As shown below, our method consistently outperforms alternatives in compression rate relative to the adjacency matrix.
>
> | Data      | Adj. Vec | Rec. Bin [3] | Graph Tok [4] | Ours     |
> | --------- | -------- | ------------ | ------------- | -------- |
> | Mol [1]   | $2.12$   | $2.16$       | $3.57$        | $9.33$   |
> | 3D PC [2] | $2.00$   | $39.46$      | $158.41$      | $572.56$ |
>
> [1] Ramakrishnan, et al. "Quantum chemistry structures and properties of 134 kilo molecules." *Scientific data,* 2014.
>
> [2] Neumann, et al. "Graph kernels for object category prediction in task-dependent robot grasping." *MLG*. 2013.
>
> [3] Dai, et al. "Scalable deep generative modeling for sparse graphs." *ICML*, 2020.
>
> [4] Chen, et al. "Graph Generative Pre-trained Transformer." *arXiv,* 2025.

---

> > ### Comment · Reviewer_4Q4T · 2025-04-04
> >
> > Thanks for the comments from the authors. Considering most of my concerns have been addressed, I would like to increase my rate.

---

> > > ### Author Response · Authors · 2025-04-05
> > >
> > > Thanks for acknowledging our rebuttal and raising the score. Feel free to let us know if you have additional questions.

---

### Official Review · Reviewer_9Exr · 2025-03-17

**Overall Recommendation:** 3

**Summary:**

This paper addresses the challenge of sustaining integrated circuit (IC) performance in the post-Moore era by proposing AnalogGenie-Lite, a decoder-only generative model for discovering novel analog circuit topologies. Leveraging lightweight graph modeling, the framework incorporates concise device-pin representations, frequent sub-graph mining, and optimal sequence modeling to significantly enhance both scalability and precision.

**Claims And Evidence:**

Yes

**Essential References Not Discussed:**

To my knowledge the most important references are discussed

**Experimental Designs Or Analyses:**

Although I did not personally execute the code, I carefully reviewed the experimental setup and the corresponding analyses, which appear to be reasonable.

**Methods And Evaluation Criteria:**

Yes

**Other Comments Or Suggestions:**

The paper is both interesting and technically solid. While the main contribution lies in enhancing scalability and thus appears incremental, these improvements meaningfully advance existing approaches. Overall, the strengths outweigh the limitations, and I recommend acceptance.

**Other Strengths And Weaknesses:**

**Strengths**
1. The paper is engaging, well-structured, and presents a computationally leaner version of the AnalogGenie Decoder.
2. While it leverages established concepts, the paper offers a meaningful contribution by combining subgraph pruning and modeling approaches to enhance scalability.

**Weaknesses**
1. Compared to the original AnalogGenie work, the innovation appears incremental. Although the improved scalability is promising, the performance gains also seem relatively modest in scope.

**Questions For Authors:**

Please see weaknesses.

**Relation To Broader Scientific Literature:**

The current paper offers a lightweight decoder version of AnalogGenie, which was recently accepted at ICLR 2025.

**Theoretical Claims:**

NA

---

> ### Author Rebuttal · Authors · 2025-03-31
>
> > **Q1:** Compared to the original AnalogGenie work, the innovation appears incremental. Although the improved scalability is promising, the performance gains also seem relatively modest in scope.
>
> We appreciate the reviewer’s insightful comments and would like to elaborate on our innovations in this paper. **Our approach is novel on three levels—graph, subgraph, and sequence—and each contributes significantly to improving circuit generation quality in terms of validity, scalability, and performance**.
>
> At the graph level, we introduce **a new graph representation that is considerably simpler than the current SOTA device-pin graph representation [1]**. Specifically, for multi-pin shared edge connections, our method achieves a space complexity of $O\left(n\right)$, compared to the $O\left(n^2\right)$ required by existing approaches. This advancement results in a 6.6% improvement in generation validity, a scalability boost of 2.9$\times$, and a performance (FoM) increase of 982.5 compared to the current SOTA [1].
>
> At the subgraph level, **we are the first, to our knowledge, to apply frequent subgraph mining techniques to an analog circuit database to identify frequently reused subcircuits and replace them with compact representations.** This novel contribution leads to an additional 7.2% improvement in generation validity, enhances scalability by 1.24$\times$, and increases performance (FoM) by 40.2 on top of the improvements achieved at the graph level.
>
> At the sequence level, we model a circuit graph as an **optimal sequence** by formulating it as the shortest closed path that visits every edge of an undirected graph at least once, effectively **solving the *Chinese Postman Problem* [2].** This approach significantly reduces sequence length and further improves generation validity by 9.7%, scalability by 1.43$\times$, and performance (FoM) by 250.2 relative to the subgraph-level method.
>
> Finally, we clarify that **our method's performance improvements are significant by comparing it to the historical development of electronic design automation algorithms for analog circuits' topology discovery**. For instance, while op-amp circuits have been extensively studied by existing EDA algorithms [3-6], **early works such as Artisan [3] achieved an 1847.7 FoM gain** using LLM-based op-amp synthesis over traditional reinforcement learning methods [4]. On top of that, the current SOTA [1] further secures an additional **975.3 FoM gain relative to Artisan**. By integrating our three innovative techniques mentioned earlier, AnalogGenie-Lite attains an overall improvement of 23.5% in generation validity, a scalability enhancement of 5.14$\times$, and a **FoM gain of 1272.9 compared to the current SOTA [1]**. These results underscore the substantial advancement our approach offers in tackling the challenging domain of analog circuit design, **pushing the circuit performance's limit in the post–Moore's law era**.
>
> [1] Gao, Jian, et al. "AnalogGenie: A Generative Engine for Automatic Discovery of Analog Circuit Topologies." *arXiv preprint arXiv:2503.00205* (2025).
>
> [2] Edmonds, Jack, and Ellis L. Johnson. "Matching, Euler tours and the Chinese postman." *Mathematical programming* 5 (1973): 88-124.
>
> [3] Chen, Zihao, et al. "Artisan: Automated operational amplifier design via domain-specific large language model." *Proceedings of the 61st ACM/IEEE Design Automation Conference*. 2024.
>
> [4] Chen, Zihao, et al. "Total: Topology optimization of operational amplifier via reinforcement learning." *2023 24th International Symposium on Quality Electronic Design (ISQED)*. IEEE, 2023.
>
> [5] Lu, Jialin, et al. "Topology optimization of operational amplifier in continuous space via graph embedding." *2022 Design, Automation & Test in Europe Conference & Exhibition (DATE)*. IEEE, 2022.
>
> [6] Zhao, Zhenxin, and Lihong Zhang. "An automated topology synthesis framework for analog integrated circuits." *IEEE Transactions on Computer-Aided Design of Integrated Circuits and Systems* 39.12 (2020): 4325-4337.

---

### Decision · Program_Chairs · 2025-05-01

**Decision:**

Accept (poster)

**Comment:**

This paper proposes AnalogGenie-Lite, a decoder-only generative framework for discovering novel analog circuit topologies. By leveraging lightweight graph modeling, frequent subgraph mining, and sequence optimization via the Chinese Postman Problem, the method improves both efficiency and generation quality. Experiments on real-world analog circuits demonstrate its effectiveness in generating diverse and high-quality designs.

All reviewers recommended accepting the paper.